# Contribution of the Tyrosinase (MoTyr) to Melanin Synthesis, Conidiogenesis, Appressorium Development, and Pathogenicity in *Magnaporthe oryzae*

**DOI:** 10.3390/jof9030311

**Published:** 2023-02-28

**Authors:** Xiaoning Fan, Penghui Zhang, Wajjiha Batool, Chang Liu, Yan Hu, Yi Wei, Zhengquan He, Shi-Hong Zhang

**Affiliations:** 1The Key Laboratory for Extreme-Environmental Microbiology, College of Plant Protection, Shenyang Agricultural University, Shenyang 110866, China; 2Key Laboratory of Three Gorges Regional Plant Genetics & Germplasm Enhancement (CTGU), Biotechnology Research Center, China Three Gorges University, Yichang 443000, China

**Keywords:** MoTyr, melanin, conidiophore stalks, appressorium, pathogenesis, *Magnaporthe oryzae*

## Abstract

Dihydroxynapthalene-(DHN) and L-dihydroxyphenylalanine (L-DOPA) are two types of dominant melanin in fungi. Fungal melanins with versatile functions are frequently associated with pathogenicity and stress tolerance. In rice blast fungus, *Magnaporthe oryzae*, DHN melanin is essential to maintain the integrity of the infectious structure, appressoria; but the role of the tyrosinase-derived L-DOPA melanin is still unknown. Here, we have genetically and biologically characterized a tyrosinase gene (*MoTyr*) in *M*. *oryzae*. *MoTyr* encodes a protein of 719 amino acids that contains the typical CuA and CuB domains of tyrosinase. The deletion mutant of MoTyr (Δ*MoTyr*) was obtained by using a homologous recombination approach. Phenotypic analysis showed that conidiophore stalks and conidia formation was significantly reduced in Δ*MoTyr*. Under different concentrations of glycerol and PEG, more appressoria collapsed in the mutant strains than in the wild type, suggesting MoTyr is associated with the integrity of the appressorium wall. Melanin measurement confirmed that MoTyr loss resulted in a significant decrease in melanin synthesis. Accordingly, the loss of MoTyr stunted the conidia germination under stress conditions. Importantly, the MoTyr deletion affected both infection and pathogenesis stages. These results suggest that MoTyr, like DHN pigment synthase, plays a key role in conidiophore stalks formation, appressorium integrity, and pathogenesis of *M*. *oryzae*, revealing a potential drug target for blast disease control.

## 1. Introduction

Melanin is a unique pigment with myriad functions and is found in various organisms, including fungi. It is produced by specialized cells called melanocytes and is responsible for the coloration of the skin, hair, and other tissues. Melanin plays multifunctional roles in an organism, including protection against environmental stresses such as ultraviolet (UV) radiation, detoxifying harmful substances, and participating in immune responses [1]. In fungi, melanin is thought to play a number of important roles in fungal defense. It protects the fungi against UV radiation and other harsh environments helping them to survive in a wide variety of environments, including on the surface of plants, in soil, and in water. Also, melanin is thought to have antioxidant properties and may help to protect fungi against oxidative stress [2]. In addition to the role of melanin in fungal defense, the substantial protective powers of melanin play decisive roles in the virulence of many fungal pathogens [3,4].

There are two main melanin synthesis pathways in fungi: Dihydroxynaphthalene (DHN) synthesis pathway and L-dihydroxyphenylalanine (L-DOPA) synthesis pathway [5,6]. The first pathway begins with malonyl-CoA as a precursor molecule, followed by a series of enzymatic modifications that leads to the polymerization of 1,8-DHN to yield DHN melanin. The second pathway begins with tyrosine or L-dopamine and leads to the dihydroxy indoles polymerization, resulting in DOPA melanin [7]. The specific pathway of melanin synthesis can vary depending on the type of fungus and the specific conditions it is experiencing. The DHN melanin resides primarily in ascomycetes and related deuteromycetes [8]. Most plant pathogens use the DHN pathway to synthesize melanin, such as *Magnaporthe oryzae*, *Aureobasidium melanogenum* and *Colletotrichum lagenarium* [9,10,11,12]. Alternatively, a few fungi synthesize melanin via the L-DOPA pathway. It is seen mostly in some animals and human pathogenic fungi, such as *Cryptococcus neoformans*, *Candida albicans* and *Histosidium capsulatum* [13,14,15].

DHN-melanin biosynthesis starts with a polyketide synthase (PKS) using acetyl coenzyme A or malonyl coenzyme A as a precursor to synthesize the polyketide chain following a series of condensation reactions (Appendix A). Once the polyketide chain has been synthesized, it is modified through a series of enzymatic reactions, including cyclization, hydroxylation, oxygenase-mediated cleavage, etc., to yield DHN-melanin. The exact sequence of reactions and the enzymes involved can vary depending on the organism in which DHN-melanin is being synthesized [16]. 

Enzymes involved in melanin synthesis have been studied in many fungi. For example, *Pks1* (encoding polyketide synthase), *Scd1* (encoding scytalone dehydratase), and *Thr1* (encoding trihydroxynaphthalene reductase) have been cloned and characterized in many fungi [10,17,18,19]. Studies have shown that deleting these genes lead to abnormal melanin synthesis and fungal dysfunction. In *Colletotrichum gloeosporioides*, deletion of the *Pks1* resulted in the failure of the mutants’ colony hyphae and appressoria to properly synthesize melanin which leads to the reduced turgor pressure of the pathogen during host attachment [20]. In *Sporothrix schenckii*, the absence of scytalone dehydratase enzyme prevents the production of DHN-melanin, resulting in its loss of dominance under various stresses [21]. In *Setosphaeria turcica*, the absence of *LAC2* (encoding laccase) blocks the development and maturation of the appressoria in mutant strains and thus decreases the pathogenicity of fungus in maize [22]. Similarly, three genes, *ALB1*, *BUF1*, and *RSY1,* were identified to be involved in *M*. *oryzae* melanin synthesis. These genes are known to be required for conidial resistance and environmental stress tolerance, and their deletion led to a loss of melanin synthesis and virulence in *M*. *oryzae* [17,23,24].

The L-DOPA pathway is another pathway for melanin synthesis in fungi. It is less common than the DHN pathway in fungi, but it is the dominant pathway for melanin synthesis in animals, including humans [25]. There are two precursor substances in the L-DOPA melanin synthesis pathway: tyrosine and dopamine. Tyrosinase (EC 1.14.18.1) is the main rate-limiting enzyme in this synthetic pathway, which is primarily involved in two reactions: catalyzing the conversion of L-tyrosine to L-dopamine and subsequent oxidation of L-dopamine to form dopamine quinone, which undergoes a series of reactions to form melanin (Appendix A) [26]. Current studies on fungal tyrosinases have mainly focused on enzyme crystal structure, its enzymatic activity and inhibitors [27,28,29,30,31]. For example, the tyrosinase gene from *Aspergillus oryzae* was cloned and expressed to achieve its application in L-DOPA biotransformation [32]. In *Armillaria ostoyae*, replacing D262 with asparagine significantly increased the catalytic efficiency of monophenolase/diphenolase [33]. However, the role of tyrosinase in fungal pathogenicity is not well understood. In *C. neoformans*, the production of L-DOPA melanin catalyzed by tyrosinase is associated with the virulence of pigmented cells. Pigmented cells of *C. neoformans* were found to be less susceptible to free radical killing and amphotericin B, a commonly used antifungal drug. They were more resistant to macrophages, which are cells of the immune system that help to protect the body from infection [34]. Overall, both DHN and L-DOPA melanin pathways are involved in fungal melanin synthesis, and melanin has been shown to play a key role in fungal pathogenesis.

Rice blast caused by *M. oryzae* is the most important and devastating fungal disease on rice, seriously affecting crop yield. The occurrence of rice blasts in a large area is mainly dependent on its infection cycle. In contrast, the normal development of appressorium, a specialized infection structure formed by *M. oryzae*, plays an important role in the pathogenicity of the rice blast fungus. The mature appressorium cell wall forms a layer of melanin that prevents glycerin and other macromolecular material leakage and plays an important role in maintaining appressorium turgor. [35]. In previous reports, the DHN melanin layer is reported as an impermeable barrier necessary for the host attachment that produces very high dilation pressure, helping the fungus to breach the plant’s cell walls and enter the plant’s tissues, which is necessary for the disease progression of *M*. *oryzae* [36]. However, the biological characteristics and pathogenic mechanism of tyrosinase, involved in the L-DOPA melanin pathway in *M*. *oryzae,* have not been reported yet. In this study, we have identified and functionally evaluated a tyrosinase gene in *M*. *oryzae* (named *MoTyr*) to determine its role in fungal development and pathogenesis.

## 2. Materials and Methods

### 2.1. Sequence Alignment Assays

The MoTyr (MGG_14598) gene amino acid sequences were acquired from the NCBI database (https://www.ncbi.nlm.nih.gov/, accessed on 2 August 2021). The tertiary structure of MoTyr protein was predicted by I-TASSER (https://zhanggroup.org/I-TASSER/, accessed on 11 October 2021). Amino acid sequences containing tyrosinase domains identified from other fungi were obtained from the NCBI database, Pfam domain prediction tool was used to predict tyrosinase domains (http://pfam.xfam.org/search/sequence, accessed on 11 October 2021). In addition, MEGAX software was used to draw a phylogenetic tree for the conserved domains.

### 2.2. Fungal Strains and Culture Conditions

*M. oryzae* strain (Y34) was used in this study. All fungal strains were cultured on complete growth medium {CM (10 g/L glucose, 2 g/L peptone, 1 g/L yeast extract, 1 g/L casamino acids, 0.5 g/L MgSO_4_, 6 g/L NaNO_3_, 0.5 g/L KCl, and 1.5 g/L KH_2_PO_4_). In the sporulation test, the RBM (rice bran medium) was used, and mycelium blocks were inoculated onto the RBM, cultured at 28 °C for 7 days in the dark, then scraped off, and cultured under light for 3 days.

### 2.3. Protoplast Preparation

Protoplasts were prepared for transformation. The wild-type strain was inoculated in CM liquid medium at 28 °C and shaken at 90 r/min for 3–4 days in the dark. The medium was filtered out, and the hyphae were fully ground in a sterile mortar, then added to the fresh CM liquid medium (containing 100 μg/mL streptomycin sulfate) and cultured in a 28 °C shaker for 90 r/min for 12–14 h. Then filtered with sterile paper and washed twice with sterilized ddH_2_O (containing 100 μg/mL streptomycin sulfate) and 1 M sorbitol (containing 100 μg/mL streptomycin sulfate). Dried the mycelium and added 40 mL lysing enzyme solution, and incubated in a 30.5 °C shaker for 2–3 h. After that, the completely lysed solution was filtered using a sterile Mira cloth and centrifuged at 5000 rpm for 10 min at 4 °C. Removed the supernatant and added 30–40 mL of Sorbitol STC, then centrifuged at 5000 rpm for 10 min at 4 °C [37]. 

Finally, the supernatant was removed again and added Sorbitol STC solution. DMSO was added (final concentration was 7%), the protoplast concentration was adjusted to 1 × 10^6^/mL, and the protoplast was separated into 300 μL volume in a 1.5 mL sterilized centrifuge tube and stored at −80 °C.

### 2.4. Targeted Gene Deletion of MoTyr Mutants

To generate the MoTyr deletion strain Δ*MoTyr*, the *MoTyr* gene was replaced by the hygromycin-resistant cassette (HPH). The primers of MoTyr-A-F/R and MoTyr-B-F/R (Appendix A) were used to amplify Y34 genomic DNA, and the upstream 862 bp (A) and downstream 849 bp (B) sequences of the *MoTyr* gene were obtained, respectively. Fragment A was fused to the pCX62 vector (after digestion with *EcoR* I and *Kpn* I), and fragment B was fused to the pCX62 vector (after digestion with *BamH* I and *Xba* I) [38]. The primers (MoTyr-A-F and HY/R; YG/F and MoTyr-B-R) amplified hygromycin overlapping fragments, respectively. For transformation, targeted gene deletion using a homologous recombination approach was used as described [39]. Then, MoTyr-ORF-F/R and UAH primers were used to verify the knockout transformants. The primers for gene deletion are listed in Appendix A.

### 2.5. Assays for Conidial Production, Growth, and Development

The strains (wild-type and Δ*MoTyr*) were cultured on RBM media at 28 °C for 7 days in the dark, then removed the hyphae and a sterile blade was used to cut out a medium and placed on a glass slide. The prepared sample was then observed under a Nikon Eclipse 80i microscope at 12, 24, 48, and 72 h. At 72 h, mycelia and conidiophore stalks were stained with lactophenol cotton blue [40]. Conidia in RBM medium were collected by adding 2 mL sterile water, and the numbers were counted under the microscope. Three biological experiments were conducted for each strain, and each experiment was repeated three times.

Conidia of the wild-type and Δ*MoTyr* were cultured on RBM media and collected to observe the germination of conidia and formation of appressoria. The conidial suspension was adjusted to 1 × 10^5^/mL and added dropwise to a hydrophobic cover slip under a microscope at 2, 4, and 6 h. Wild type and mutant spores were exposed to −20 °C and UV for 12 h and 30 min, respectively, and observe the germination of spores. Three biological experiments were conducted for each strain, and each experiment was repeated three times.

### 2.6. Analysis for Appressorium Integrity 

In order to test the effect of MoTyr on the integrity of the appressorium wall and turgor pressure generated by it, 50 μL of spore suspension with a concentration of 5 × 10^4^ conidia/mL was dropped onto the surface of the hydrophobic membrane and incubated at 28 °C for 24 h. After the appressorium was generated, the water layer covering the spore was carefully removed. It was treated with 50 μL glycerol solution with concentrations of 1 M, 2 M, 3 M and 4 M, respectively, and kept wet at 28 °C for 10 min. The appressorium collapse was observed under a microscope and recorded [41].

Under the same experimental conditions, 20%, 25%, 30% and 35% PEG3350 solution instead of glycerol was used to treat the appressorium, and the collapse of appressoria cells was observed by microscope. Each strain of the above experiments was used to observe the appressoria isolated from the plasma wall, and the appressoria number was at least 100. The experiment was repeated 3 times.

### 2.7. Rice Sheath Penetration and Plant Infection Assays

To test the pathogenicity of the *MoTyr*, the wild-type and Δ*MoTyr* were cultured on RBM and collected conidia as previously described. At the four-leaf stage of rice seedlings (*Oryza sativa* cv. Lijiangxintuanheigu), the 1 mL spore suspension (1 × 10^5^ conidia/mL in 0.25% gelatin) was inoculated on the surface of rice leaves and incubated in the dark for 24 h in a culture chamber at 28 °C, then changed to a photoperiod of 16 h for 7 days. For the infection test of barley leaves, 20 μL spore suspension was inoculated on intact and injured barley leaves and cultured under similar conditions as described above. 

The spore suspension (1 × 10^5^ conidia/mL) was injected into the leaf sheath of rice and then put in a moist environment. The infection rates of leaf sheaths and the types of infectious hyphae (IH) at 12, 24 and 48 h post-inoculation (hpi) were analyzed and repeated in triplicate as previously described. The experiment performed three biological replicates.

### 2.8. Extraction and Purification of Melanin

To determine the melanin content of the wild-type and Δ*MoTyr*, collected conidia suspensions (1 × 10^5^ conidia/mL) were cultured at 28 °C for 24 h for appressoria production. Centrifuged at 5000 rpm for 10 min, the supernatant was dumped, the precipitate was dried and weighed, and 1 mL of 1 M NaOH was added (according to 1:120 (*w*/*v*)). Continue heating at 121 °C for 20 min to extract melanin; after extraction, the supernatant was filtered, and the volume was fixed to 1 mL, with 1 M NaOH as a blank control. The absorbance value was measured at 405 nm.

### 2.9. Statistical Analysis

All the experiments were performed at least three times. SPSS Statistics 25 was used for data statistics and analysis. The analysis was performed using an independent samples t-test. Error bars represent the standard deviation. * indicates a statistically significant difference at *p* < 0.05. ** indicates a highly significant difference at *p* < 0.01.

## 3. Results

### 3.1. MGG_14598 Encodes a Tyrosinase in M. oryzae 

Tyrosinase exists in many species, and its functional mechanisms have been reported in many species [33,42,43,44]. At present, the mechanism of tyrosinase action in blast fungus is not clear. 

A putative tyrosinase, MGG_14598, was identified in the *M*. *oryzae* genome that encodes a putative tyrosinase protein and contains a common central domain of tyrosinase in *M*. *oryzae*. Hence, the gene was named *MoTyr*, with a 2160 bp open reading frame length, encoding a protein of 719 amino acids. Tyrosinase is present in many fungi. We compared the relationship between tyrosinase proteins of six fungi and *M. oryzae* MoTyr. MoTyr was closely related to tyrosinase in *Neurospora crassa* (Figure 1A). MoTyr protein structure has an important domain: the Tyrosinase domain (79–322) (Figure 1B), and has two copper coordination sites, CuA (83–122) and CuB (277–315), and six conserved histidine residues that are involved in the coordination between the two copper sites [45,46]. Consistent with our results, alignment comparison and 3D structure revealed that MoTyr protein had conserved histidine residues with *A. ostoyae* and *Colletotrichum spinosum* at residues H88, H112, H121, H283, H287 and H310 (H88, H112, H283 for CuA and H283, H287, H310 for CuB; Figure 1B,C). These results indicate that domains of tyrosinase are well conserved in the fungal group.

To further evaluate the biological characteristics of the *MoTyr* gene, we analyzed the expression abundance of *MoTyr* at different developmental stages of *M*. *oryzae*. We found that the expression of *MoTyr* was downregulated on the 5th and 6th day of mycelia growth. On the 7th day of development, the mycelium was removed and developed into conidiophore stalks. However, the expression of *MoTyr* was up-regulated in the conidiophore stalks during 6–12 h of development. The expression of this gene began to be downregulated in the later stages of development (Figure 1D). In addition, we injected wild-type spore suspensions into rice leaf sheaths and examined the expression of *MoTyr* at different infection stages. The results showed that *MoTyr* was highly expressed at the early stage of infection (12 and 24 hpi) but was downregulated at 36 hpi of infection (Figure 1E). These results suggest that *MoTyr* may be involved in conidiophore stalks development and may play a key role in the early stage of infection. 

### 3.2. MoTyr Is Important for Conidiophore Stalks and Conidiation Development

The expression of *MoTyr* was high during the development of the conidiophore stalks of *M*. *oryzae*, so we speculated that *MoTyr* might be involved in the development of the conidiophore stalks. To prove our hypothesis, we deployed a homologous recombination approach for targeted gene deletion of the *MoTyr* gene using the pCX62 vector. Two *MoTyr* gene deletion strains (Δ*MoTyr-10*, Δ*MoTyr-31*) were successfully obtained and further confirmed through PCR (Appendix A).

We shaved off the hyphae on RBM, and a piece of the culture medium was cut with a sterile blade and cultured at 28 °C and observed the development state of conidiophore stalks at 12, 24, 48 and 72 h under a microscope. The results showed that the development of conidiophore stalks of the mutants was slower, and the development of mycelia was sparse compared with that of the wild-type at the same development time (Figure 2A). We used lactophenol cotton blue to stain the conidiophore stalks, and the results showed that the number of conidiophore stalks in the mutant was significantly lower than wild-type (Figure 2B). 

The conidiation may be affected by the obstruction of conidiophore stalks formation. We examined the conidial production of the mutants and wild-type and found that the spores produced by the knockout mutant Δ*MoTyr* were significantly reduced than that of the wild-type (Figure 2C). The normal development of conidia and appressorium is an important factor in the pathogenicity of *M. oryzae*. The results showed that deleting MoTyr did not affect mutants’ vegetative growth and normal spore morphology development (Appendix A). In addition, the spore germination and appressorium formation of the MoTyr deletion mutants were not significantly different from the wild-type (Appendix A). Our results show that the deletion of MoTyr has not affected the conidial germination and appressorium morphogenesis, but it has reduced the number of conidiophore stalks and spores, which indicates that MoTyr is important for conidiophore stalks and conidiation formation.

### 3.3. MoTyr Is Essential for the M. oryzae Appressoria Integrity 

Tyrosinase is a rate-limiting enzyme involved in melanin synthesis. The appressorium of *M. oryzae* is the key factor in infecting the host plant, and the normal formation of melanin plays an important role in maintaining the appressorium function of *M. oryzae*. We test the effect of deletion of the *MoTyr* gene on the cell wall integrity of the appressorium of *M. oryzae*, the appressoria formed by wild-type and mutant strains on the hydrophobic membrane were treated with glycerol and PEG solution. Our results showed that cell collapse occurred in all strains after the appressoria were treated with glycerol at different concentrations. The degree of appressoria collapse of Δ*MoTyr* was stronger than that of the wild-type under the treatment of 1 M, 2 M and 3 M of glycerol (Figure 3A,B). The appressoria collapse rate of Δ*MoTyr-10/*Δ*MoTyr-31* was 1.58/1.54, 1.43/1.72, and 1.10/1.24 times that of the wild-type at 1 M, 2 M and 3 M concentrations of glycerol, respectively. However, there was no significant difference in the collapse rate between the mutants and wild-type under the treatment of 4 M of glycerol, indicating that the concentration of glycerol had reached the maximum tolerance of the appressoria (Figure 3B). At the same time, the appressoria collapsed after treatment with different concentrations of PEG solution, and the appressoria collapse rate of the MoTyr deletion strains was significantly higher than that of the wild-type (Figure 3C,D). These results suggest that MoTyr is important for the cell wall integrity of the appressorium.

### 3.4. MoTyr Loss Affected Melanin Synthesis and Stress Resistance in M. oryzae

Appressorium turgor is generated by the rapid influx of water into the cell against a concentration gradient of glycerol that is maintained in the appressorium by a specialized, melanin-rich cell wall. The normal synthesis of melanin plays an important role in maintaining appressorium turgor. In our study, we tested tyrosinase activity in the hyphae of the mutant and wild-type and found that the tyrosinase activity in the mutant was significantly lower than that of the wild-type (Figure 4A). The results showed that the *MoTyr* gene is involved in the synthesis of tyrosinase. Tyrosinase is an important rate-limiting enzyme in the L-DOPA melanin synthesis pathway, but whether the deletion of this gene will affect melanin synthesis in *M*. *oryzae* is unknown. We tested melanin levels in both mutants and wild-type; the marginal hyphae of mutant and wild-type were inoculated into CM liquid medium after 7 days of shock culture at 28 °C and found that the blackening level of the mutants was weaker than that of the wild-type (Figure 4B). In addition, compared to the variation of melanin content in conidia and appressoria, the results showed that the melanin content in the mutant strains was significantly lower than that of the wild-type (Figure 4C). These results suggest that MoTyr is involved in melanin synthesis and plays an important role in the appressorium development of *M*. *oryzae*.

In fungi, normal synthesis of melanin favors increased fungal tolerance to abiotic and biological environmental stresses, including radiation, extreme temperatures, dryness, metal toxicity, and host immune responses [47,48,49]. We examined the spore germination of the MoTyr deletion mutant under stress conditions. After freezing at −20 °C for 24 h, the conidial germination rates of the Δ*MoTyr* at different times were significantly lower than those of the wild-type (Figure 4D). After exposure to UV for 30 min, the conidial germination rate of Δ*MoTyr* strains was significantly lower than that of the wild-type (Figure 4E). These results suggest that the deletion of MoTyr inhibits melanin synthesis, thereby reducing the ability of *M. oryzae* to resist abiotic and biological environmental stresses.

### 3.5. MoTyr Is Important for Pathogenicity in M. oryzae

In order to detect the role of MoTyr in the pathogenic development of *M. oryzae*, we collected conidia from wild-type and Δ*MoTyr* strains and tested them for pathogenicity. When intact susceptible rice seedlings were spray-inoculated with conidial suspension, we found that the rice leaves inoculated with mutant spores were less susceptible to the disease and had lower disease incidence than those inoculated with wild-type (Figure 5A). Similarly, when drop-inoculation was assayed, the mutant strains still showed weaker pathogenicity than the wild-type (Figure 5B). In addition, we treated the intact and injured barley leaves with wild-type and Δ*MoTyr* hyphae and found that under intact treatment, the disease lesions on mutant strains’ inoculated barley leaves were smaller than wild-type (Figure 5C), but on injured leaves, the degree of disease was similar to that of the wild-type (Figure 5D). These results suggest that MoTyr plays an important role in pathogenesis, and it may be that the loss of MoTyr resulted in decreased infection ability of appressorium and thus reduced its pathogenic capability. 

The collapse of the appressorium would result in the weak infection ability and hyphal development of *M. oryzae* in host cells. Therefore, we conducted a microscopic observation of rice leaf sheaths infected by conidia. At 12 hpi, many mature (black) appressorium were formed in both the wild-type and the mutant, but appressorium formation in the mutant was less than in the wild-type. After inoculation for 24 h, the wild-type infected mycelia could develop normally in rice cells and spread to the whole cell. Still, the primary infectious hyphae were just formed in Δ*MoTyr*. At 48 hpi, the wild-type infected mycelium had begun to spread to neighboring cells, and the mutants had also spread to the entire cell (Figure 5E). In order to quantitatively compare the infection of rice leaf sheath cells by different strains, we divided the infection types into three categories (Type Ⅰ: mature appressoria; Type Ⅱ: formed primary hyphae and invasive hyphae extended and branched in one cell; Type Ⅲ: invasive hyphae crossing to neighboring cells) and analyzed different infection types of more than 50 germinated conidia after inoculation. At 12 hpi, about 8.81% of invasive and primary hyphae had formed in the wild-type, whereas only 0.97% and 3.10% of invasive and primary hyphae formed in the two mutants, respectively (Figure 5F). At 24 hpi, the wild type had formed type III infective hyphae, which can expand to adjacent cells; However, in the mutants, hyphae only developed to type II. At 48 hpi, in the wild type, the infection hyphae formation rate of type III had reached about 21%; At this time, the mutant also formed type III hyphae, but the formation rate only reached 5.26% and 2.00%, respectively (Figure 5F). These results indicated that MoTyr plays an important role in maintaining the infectious development of *M. oryzae*.

## 4. Discussion

Melanins are insoluble pigments that contribute to microorganism survival, colorization, and UV protection and are also related to fungal virulence in plant and human hosts [25,47]. Melanin has two main synthesis pathways in fungi: the DHN and L-DOPA pathways [5,6]. Various studies have been done on DHN pathway-related enzymes, and their role in pathogens’ development and virulence [10,17,18,19,20,21,22,23,24]; however, the L-DOPA pathway has yet to be studied in detail, especially in fungal pathogens. Tyrosinase is the main rate-limiting enzyme involved in the synthesis of L-DOPA melanin, which widely exists in animals, plants, microorganisms, and the human body [32,44,50,51]. In recent years, the research on tyrosinase mainly focuses on medicine, beauty, food and other fields, but there are no relevant studies on tyrosinase in pathogenic fungi. 

Rice blast, caused by pathogenic fungi *M. oryzae*, is an important fungal disease of rice [52,53]. It has been reported that the loss of rice yield caused by *M. oryzae* is enough to feed 60 million people every year [54]. Fungal melanin is considered an important virulence factor in a number of fungal species, acting as non-specific armor during infection that protects the fungus against the host immune system [55,56]. In *C*. *neoformans*, the production of L-DOPA melanin is related to the toxicity of pigmentation cells [34]. Tyrosinase is involved in the L-DOPA melanin synthesis pathway, but its role in the pathogenesis of *M*. *oryzae* remains unknown. Here, we identified and deployed cell biology, functional genetics, and biochemical techniques to functionally evaluate the contribution of the uncharacterized gene (MGG_14598) named tyrosinase (MoTyr) in morphological and pathogenic development of the economically destructive rice blast fungus. Tyrosinase orthologs are well-conserved in many fungi [32,33]. Our study showed that MGG_14598 encoded protein and all the fungal tyrosinase proteins characterized so far share a conserved tyrosinase domain, two copper coordination sites, CuA and CuB and six conserved histidine residues. We also demonstrated that MoTyr shared a closer phylogenetic lineage with *N*. *crassa*. These results are consistent with the reported structural characteristics of fungal tyrosinase, which confirms that MGG_14598 encodes a tyrosinase protein. The MoTyr expression pattern was observed to be upregulated during the stalk formation and early stage of infection; these results indicate that MoTyr may be involved in the pathogenic development of *M*. *oryzae*. 

In nature, *M*. *oryzae* mainly uses asexual spores, conidia, as a source of infection to complete its entire infection cycle. The normal development and formation of conidia is an essential part of the pathogenicity of *M*. *oryzae* [57]. To understand whether MoTyr is involved in the formation and development of conidia in *M*. *oryzae*, we generated Δ*MoTyr* mutant strains by deleting the MoTyr gene in wild-type *M. oryzae* strain, Y34. The results showed that the conidia morphology was not different from that of the wild-type. Still, the number of conidia of the mutant strain was significantly reduced compared with the wild-type. In previous studies, researchers have cloned and identified many genes that control sporulation. For example, the conidia morphology of the deleted mutants of *CON1*, *CON2* and *CON4* genes related to sporulation was abnormal, and the ratio of sporulation to wild-type also decreased significantly [58]. Similarly, deletion of *MoSTU1*, *MoHOX2* and *MoCON7* genes can lead to abnormal conidia formation in *M*. *oryzae* [59]. 

The development of conidiophore stalks plays a key role in the normal formation of conidia. The deletion of MoTyr resulted in conidia decrease in the number of *M*. *oryzae*. To further understand whether the decrease of conidia was related to the abnormal formation of conidiophore stalks, we stained the conidiophores with lactate phenol cotton blue. Our results showed that the mutant formed fewer conidiophore stalks than the wild type. This result is the same as *COS1*, a conidiophore stalk-less1 gene; the deletion of *COS1* results in a complete loss of conidiophore stalks development, resulting in the inability to produce conidia in *M*. *oryzae* [60]. Our results are identical to the effect of the carbonic anhydrase (*CA1*) gene; the deletion of CA1 resulted in a decrease in the number of conidiophore stalks and conidiation [61]. Furthermore, the deletion of glycogen synthase kinase (GSK1) in *M*. *oryzae* results in the complete loss of conidiophore stalks and conidiation [62]. These results indicate that the abnormal development of conidiophore stalks often affects the formation of conidia. But these results differ from the role of *HTF1*, as the deletion of *HTF1* blocked conidiation but not the development of conidiophore stalks in *M*. *oryzae* [63]. This result suggests that the formation of conidiophore stalks may not be directly regulated by *HTF1* but can affect conidia formation. However, in our study, the number of conidiophore stalks and conidia decreased in the MoTyr deletion mutants, indicating that the MoTyr may directly regulate the formation of conidiophore stalks and then regulate the normal development of conidia.

In fungi, melanin increases the fungal tolerance towards extreme environments, including radiation, oxidants, enzymatic lysis, extreme temperatures, toxic metals and ionizing radiation and host immune responses [24,47,64]. In *Pestalotiopsis fici*, the deletion of PfMAE led to increased UV sensitivity of spores [65]. At extreme temperatures (42–47 °C), *Streptomyces galbus* produces more melanin, which thickens its cell wall and protects it from heat damage [66]. In *C*. *neoformans*, melanin also buffers physical damage, such as cold temperatures and protects cells from intracellular and extracellular ice crystals [67]. In *M*. *oryzae*, *ALB1*, *RSY1,* and *BUF1* are essential for conidial resistance to UV exposure, oxidation and freezing damage: deletion of these genes inhibited conidial germination [24]. Comparable results were found in our study, where the deletion of MoTyr led to reduced melanin synthesis and inhibited conidial germination under UV and freezing stress.

Melanin is produced by many organisms, including pathogenic fungi, where it can play a role as an important virulence factor. The appressorium is a highly pigmented and thick-walled structure formed by *M. oryzae* during infection. It is a site of intense melanin production and an important intermediate organ in the host infected by *M*. *oryzae*. In the process of appressorium maturation, glycerol and other substances continue to accumulate in the appressorium up to 3.0 MPa, making its internal turgor constantly rise to about 8.0 MPa, providing enough internal pressure for it to penetrate the host cuticle cells [68]. Our study found that the appressorium of the Δ*MoTyr* mutant strains was more prone to collapse under different concentrations of glycerol and PEG. Like our study, appressorium turgor pressure was abnormal after MoSwi6 (an APSES family transcription factor) deletion under glycerol treatment, and most of the appressorium could not penetrate rice cells [69]. However, in the corn pathogen *Colletotrichum graminicola*, deletion of *CgPKS1* led to non-melanized cell walls but normal turgor pressure in its appressoria [70]. The abnormal synthesis of melanin leads to a low-integrity appressorium wall, which was unable to produce high appressorium turgor to penetrate the plant [36]. In *M*. *oryzae*, MoCA1 participates in appressorium maturation and morphogenesis; it was found that the loss of the gene reduced melanin synthesis and infection ability [61]. C3HC type Zinc-finger protein (ZFC3) and deletion of this gene led to increased melanin synthesis and its ability to infect rice cells [71]. These results suggest that *M*. *oryzae* requires melanin for appressorial turgor. These results are similar to our study, where the loss of tyrosinase leads to a lower level of melanin synthesis and causes the appressorium to collapse more easily. Therefore the infection ability of mutant strains decreased significantly. Also, during the development of appressorium, melanin synthesis was blocked, and the normal turgor pressure could not be maintained, which resulted in a weakened penetration ability of the penetration peg.

Previous studies have shown that *M*. *oryzae* utilizes the DHN melanin pathway. In our study, we found that after the deletion of MoTyr, melanin synthesis in *M*. *oryzae* decreased significantly. To this end, we detected the expression levels of *ALB1*, *BUF1* and *RSY*, which are key genes involved in the melanin synthesis pathway of *M*. *oryzae*. The results showed that the expression level of *BUF1* and *RSY* was significantly decreased in mutant strains (Figure 6). However, whether tyrosinase is also involved in melanin synthesis in *M*. *oryzae* was unknown. Therefore, we hypothesized that the deletion of tyrosinase affects the expression of genes associated with the DHN melanin synthesis pathway, possibly because tyrosinase has a catalytic substrate in *M*. *oryzae*. However, this substrate may have a feedback regulatory relationship with the substrates of the DHN melanin synthesis pathway, which leads to a decrease in the expression level of genes related to the DHN melanin synthesis pathway.

In conclusion, we demonstrate that MoTyr is required for conidiogenesis, appressorium development and melanin synthesis, which leads to the successful pathogenic development of *M*. *oryzae*. Tyrosinase is a rate-limiting enzyme involved in DOPA melanin synthesis, but the mechanism of its involvement in the DHN melanin synthesis pathway of *M*. *oryzae* remains unclear and needs further study.

## Figures and Tables

**Figure 1 jof-09-00311-f001:**
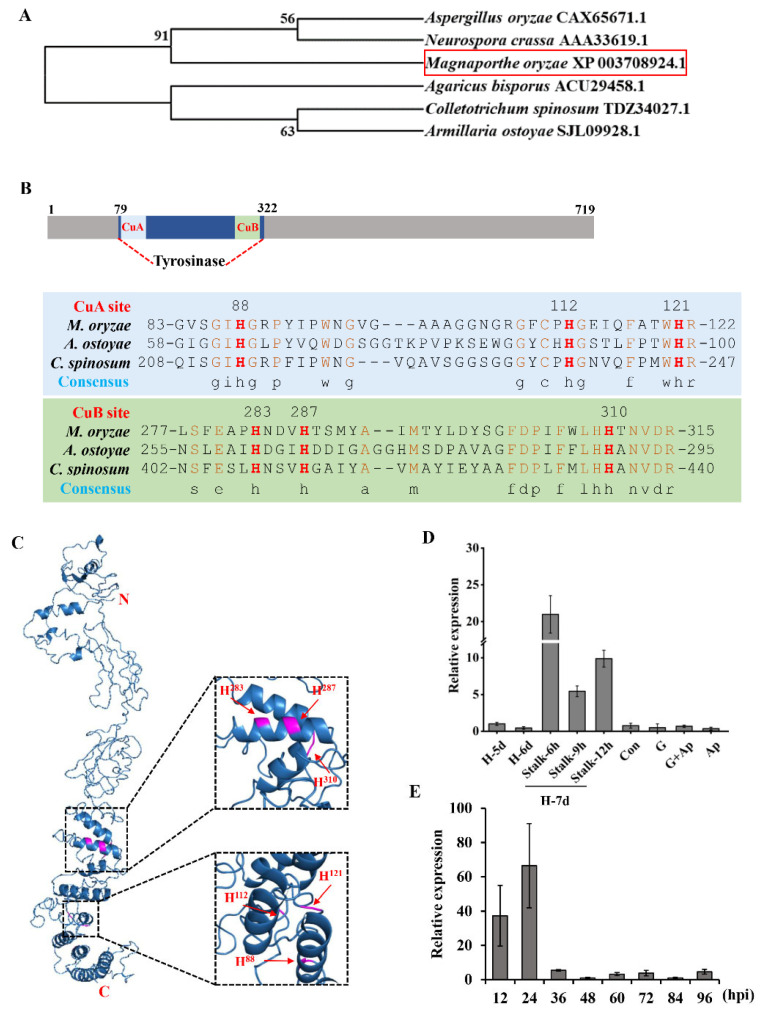
Phylogenetic analysis, protein structure, domain alignment, and expression of tyrosinase MoTyr in *M*. *oryzae*. (**A**) Phylogenetic tree constructed with tyrosinase homologs from *M*. *oryzae*, *A*. *oryzae*, *N*. *crassa*, *A*. *bisporus*, *C*. *spinosum*, and *A*. *ostoyae*. (**B**) Amino acid sequence alignment of tyrosinases from *M*. *oryzae*, *A*. *ostoyae,* and *C*. *spinosum*. Histidine residues participating in copper coordination are marked in bold red font. (**C**) Tertiary structure (3-D) of MoTyr. (**D**) Expression level of *MoTyr* in *M*. *oryzae* at different developmental stages. (**E**) Expression level of *MoTyr* in leaf sheath infected at different periods in wild-type. H, hyphae; Stalk, conidiophore stalks; Co, conidia; G, germ tube; Ap, appressoria.

**Figure 2 jof-09-00311-f002:**
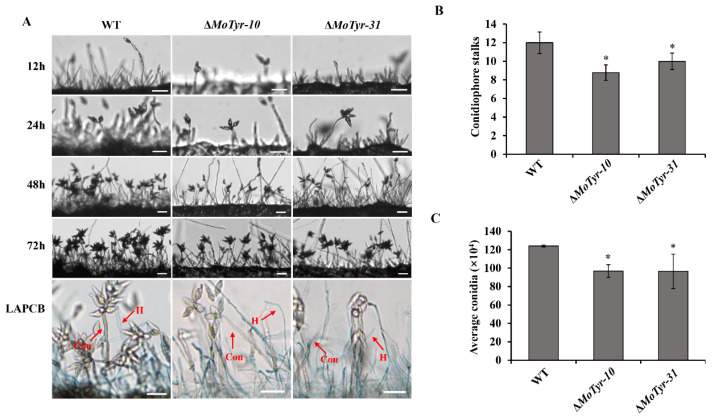
Conidiophore stalks and conidiation development of wilt-type and Δ*MoTyr* mutant stains. (**A**) The conidiophore stalks of the wild-type and Δ*MoTyr* strains were induced for 12, 24, 48, and 72 h; strains were stained with lactophenol cotton blue at 72 h. The hyphae were stained blue, whereas the conidiophore stalks and conidia were grey; Scale bar = 20 μm; LAPCB, Lactophenol cotton blue; H, hyphae; Con, conidiophore stalks. (**B**) Statistical analysis of conidiophore stalks in wild-type and Δ*MoTyr* strains. (**C**) Conidia count in wild-type and Δ*MoTyr* strains. Asterisks * represent a statistically significant difference of *p* < 0.05. Error bars indicate the mean ± SD from three independent experiments.

**Figure 3 jof-09-00311-f003:**
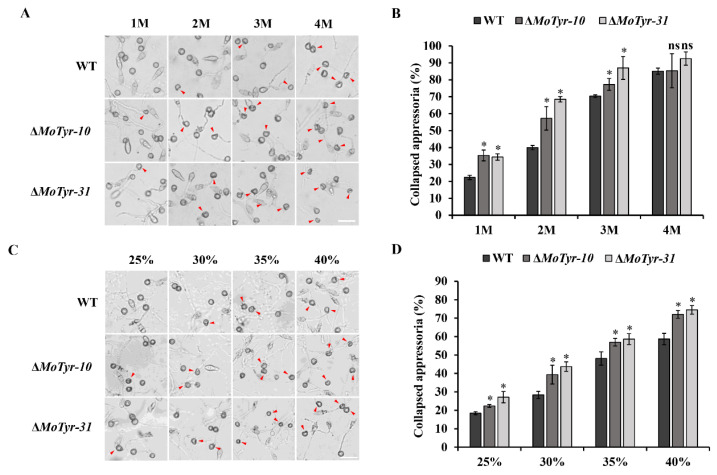
Appressoria development analysis of the wild-type and Δ*MoTyr* mutant strains. (**A**) Cytorrhysis assay for appressorium turgor pressure determination. Conidia (1 × 10^5^ conidia/mL) of wild-type and mutant strains were seeded on hydrophobic mulch, and appressoria produced after 24 h of dark culture at 28 °C were treated with different concentrations (1, 2, 3, and 4 M) of glycerol; The red arrow pointed to the collapsed appressorium; Scale bar = 20 μm. (**B**) A statistical representation of proportion-collapsed appressoria recorded for the individual strains under the different concentration glycerol solutions treatments. (**C**) Appressorium turgor pressure determination was treated with different concentrations (25%, 30%, 35%, and 40%) of PEG; The red arrow pointed to the collapsed appressorium; Scale bar = 20 μm. (**D**) A statistical representation of proportion-collapsed appressorium recorded for the individual strains under the different concentration PEG treatments. For each biological replicate of Figure A and C, a total of 100 appressoria were counted (*n* = 100 × 3). Asterisks * represent a statistically significant difference of *p* < 0.05; ns represent not significant.

**Figure 4 jof-09-00311-f004:**
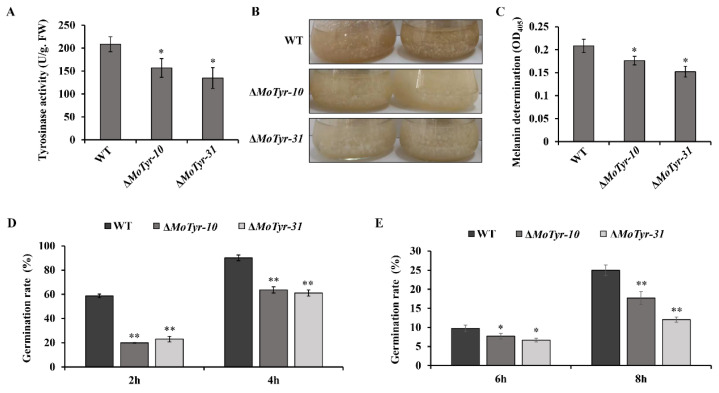
Detection of melanin levels in mutant strains and conidial germination of *M. oryzae* spores under environmental stresses. (**A**) Detection of tyrosinase activity in hyphae of mutants and wild-type. (**B**) Melanin was decreased in the Δ*MoTyr* mutant strains. (**C**) The melanin content of strains in conidia and appressoria. (**D**) Germination rate of conidial at −20 °C for 12 h. (**E**) Germination rate of conidial exposure to UV for 30 min. The analysis was performed using an independent samples t-test. Note: Statistical analyses were performed with data obtained from three biological replications with three technical replicates each time. Asterisks * represent a statistically significant difference of *p* < 0.05, and ** represent a statistically significant difference of *p* < 0.01.

**Figure 5 jof-09-00311-f005:**
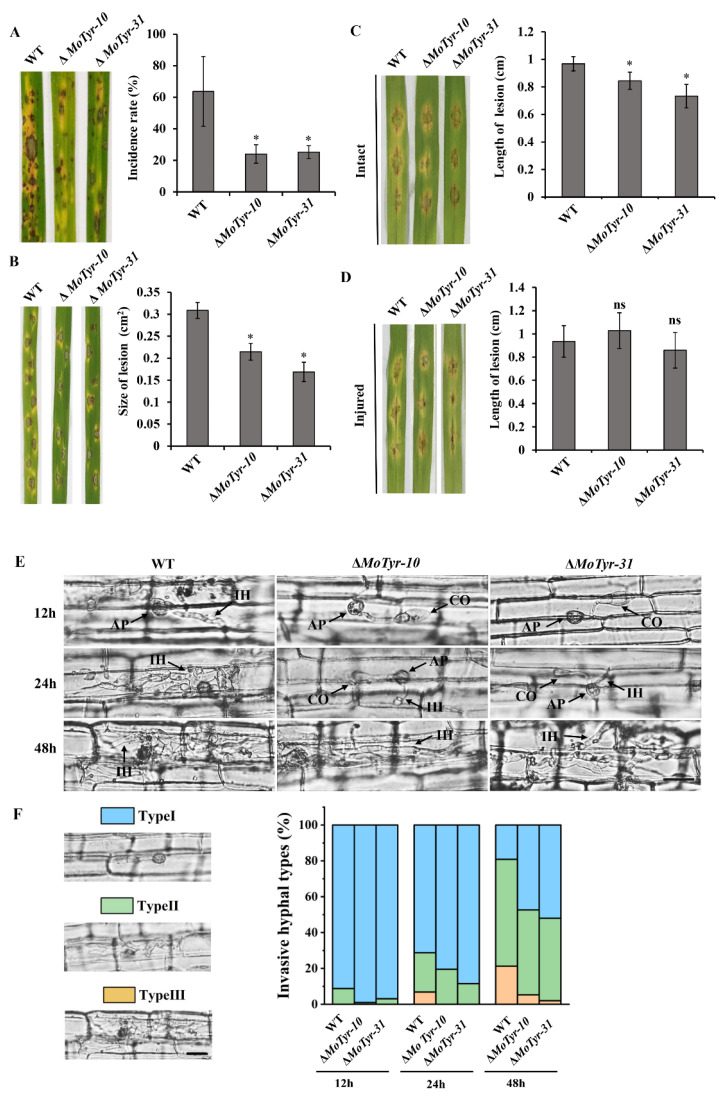
Pathogenesis analysis of the wild-type and mutant strains. (**A**) Spray-inoculation assay. Disease symptoms at 5 dpi of leaves by spraying with conidia (1 × 10^5^/mL). The graph shows the statistics of the incidence rate of rice leaves. (**B**) Drop-inoculation on leaves. Disease symptoms following the inoculation of rice leaves with 10-mL droplets of conidia (1 × 10^5^/mL). Representative leaves were photographed at 5 dpi. The graph shows the statistics of lesion size. (**C**,**D**) Represent the hyphae-mediated blast lesions on intact and injured barley leaves inoculated with mycelial plugs from the wild-type and Δ*MoTyr* mutant strains. The graph showed the statistics of the length of the lesion on the barley leaves. (**E**) Rice leaf sheath infection assay. The spore suspensions of wild-type and mutant strains were injected into rice leaf sheaths and cultured at 28 °C. The infection of rice leaf sheaths was observed for 12 h, 24 h and 48 h. The representative infection types were photographed. IH, infectious hyphae, AP, appressoria and CO, Conidia; Scale bar = 10 μm. (**F**) Statistics of different infection types. Type I: mature appressoria; Type II: formed primary hyphae and invasive hyphae extended and branched in one cell; Type III: invasive hyphae crossing to neighboring cells. Error bars indicate the mean ± SD from three independent experiments. ns represents not significant, and asterisks * represent a statistically significant difference of *p* < 0.05.

**Figure 6 jof-09-00311-f006:**
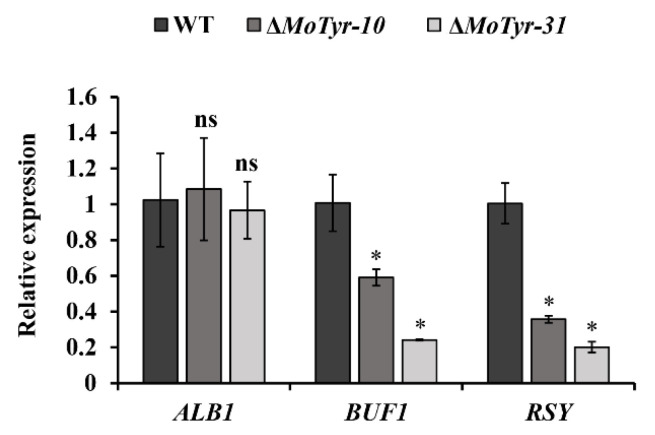
Expression analysis of melanin-related genes *ALB1*, *BUF1* and *RSY1*. Asterisks * represent a statistically significant difference of *p* < 0.05; ns represent not significant.

## Data Availability

Not applicable.

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
