# Peer review of "Contribution of the Tyrosinase (MoTyr) to Melanin Synthesis, Conidiogenesis, Appressorium Development, and Pathogenicity in Magnaporthe oryzae"

_jof, 2023, doi:10.3390/jof9030311_

Round 1

Reviewer 1 Report

The manuscript is very well written and the results are well presented. I recomend this work for its publication

minor observations

separate numbers from units for example  12h to 12 h, 3M to 3 M etc revise all the manuscript.

just revise to make the statement congruent. is affected or not?

In addition, the SPORE GERMINATION and appressorium formation of the MoTyr deletion mutants WERE NOT significantly different from the wild-type (Supplement Figure 4). Our results shows that deletion of MoTyr has not only affected the CONIDIAL GERINATION and appressorium formation, but it has also reduced the formation  of conidiophore stalks and sporulation, which indicated that MoTyr is important for conidiophore stalks and conidiation development.

Author Response

Thankyou for keen observation and valuable suggestions. All the comments have been reponded. Kindly find the attached file below. The revised manuscript will be uploaded soon.

Reviewer 2 Report

Dear Authors,

The manuscript “Contribution of the Tyrosinase (MoTyr) to melanin synthesis, conidiogenesis, appressorium development, and pathogenicity in Magnaporthe oryzae” characterizes genetically and biologically a tyrosinase gene (MoTyr) in M. oryzae. The deletion mutant of MoTyr ((ΔMoTyr) was obtained.

The materials and methods used are correct. The results showed that conidiophore stalks and conidia formation of M. oryzae was significantly reduced in ΔMoTyr.

In my opinion the manuscript can be Accept in present form

Author Response

Thankyou for the kind comments. We appreciate the reviewer consideration for our paper acceptance.

Reviewer 3 Report

The manuscript "Contribution of the Tyrosinase (MoTyr) to melanin synthesis, conidiogenesis, appressorium development, and pathogenicity in Magnaporthe oryzae" presents interesting research results but requires minor revisions before publication.

Detailed notes:

line 52-55 - add information on Aureobasidium sp., which biosynthesizes melanins and can be used industrially.

line 59 - I would suggest moving very nice drawings from Supplement to the main text (applies to all drawings).

The primer sequences should be given in the methodology.

Chapter 2.8 - were the melanins purified, was the absorbance measured in the supernatant after extraction? Please describe the methodology in more detail.

The Statistical analysis chapter should not be a separate chapter, but a subchapter of the methodology. Besides, the statistical analysis should be described in more detail (what tests were used)?

In the discussion, the authors do not refer to the latest literature (only about 25% of the literature cited by the authors in the manuscript is from the last 5 years).

In item 21 of the list of literature, the year of publication is missing.

Author Response

We appreciate the reviewer's positive feedback.  All the suggestions and comments have been responded to in the file attached below. The corrected version of the manuscript will be updated soon.

Reviewer 4 Report

In this study the authors present the results concerning tyrosinase in relation to melanin synthesis, in the fungus M. oryzae model.
I have some suggestions.
1) I would remove the final part of the introduction, from "In this study...." or from "Our results....." to the end, as I believe it is not appropriate to summarize the results in the introduction
2) regarding the figures: the asterisks represent the statistical difference of the mutant strains compared to the WT strain, right? Why put lines? Can't they be eliminated? I don't consider them necessary.

Author Response

(The authors gave the same response as above.)
